

# Construction and validation of co-expression vector for rice alpha tubulin and microtubule associated protein respectively fused with fluorescent proteins

Chenshan Xu[1], Xiaoli Zhu[1], Aihong Xu[2], Jian Song[1] and Shuxia Liang[1]

[1] College of Life Science, Dezhou University, Dezhou, Shandong, China
[2] College of Ecology, Resources and Environment, Dezhou University, Dezhou, Shandong, China

Corresponding author
Chenshan Xu, michael_10@163.com

## ABSTRACT

Microtubule (MT) consists of α-tubulin and β-tubulin. The dynamic instability regulated by various microtubule associated proteins (MAPs) is essential for MT functions. To analyze the interaction between tubulin/MT and MAP *in vivo*, we usually need tubulin and MAP co-expressed. Here, we constructed a dual-transgene vector expressing rice (*Oryza sativa*) α-tubulin and MAP simultaneously. To construct this vector, plant expression vector pCambia1301 was used as the plasmid backbone and Gibson assembly cloning technology was used. We first fused and cloned the *GFP* fragment, *α-tubulin* open reading frame (ORF), and *NOS* terminator into the vector pCambia1301 to construct the p35S::GFP-α-tubulin vector that expressed GFP-α-tubulin fusion protein. Subsequently, we fused and cloned the *CaMV 35S* promoter, *mCherry* fragment, and *NOS* terminator into the p35S::GFP-α-tubulin vector to generate the universal dual-transgene expression vector (p35S::GFP-α-tubulin-p35S::mCherry vector). With the p35S::GFP-α-tubulin-p35S::mCherry vector, *MAP* ORF can be cloned into the site of 5′ or 3′ terminus of *mCherry* to co-express GFP-α-tubulin and MAP-mCherry/mCherry-MAP. To validate the availability and universality of the dual-transgene expression vector, a series of putative rice *MAP* genes including *GL7*, *OsKCBP*, *OsCLASP*, and *OsMOR1* were cloned into the vector respectively, transformed into *Agrobacterium tumefaciens* strain, and expressed in *Nicotiana benthamiana* leaves. The results indicated that all of the MAPs were co-expressed with α-tubulin and localized to MTs, validating the availability and universality of the vector and that GL7, OsKCBP, OsCLASP, and OsMOR1 might be MAPs. The application of the co-expression vector constructed by us would facilitate studies on the interaction between tubulin/MT and MAP in tobacco transient expression systems or transgenic rice.

## INTRODUCTION

Microtubules (MTs) are ubiquitous cytoskeletal filaments critical for multiple cellular processes, including cell morphology, cell division, and intracellular trafficking (*Alushin et al., 2014*). These hollow cylinders are built by the self-association of αβ-tubulin dimers

that are composed of α- and β-tubulin subunits. The α- and β-tubulins are both globulins, and both exist in several isotype forms and undergo a variety of posttranslational modifications (*Ludueña, 1998*). The α- and β-tubulins share 40% amino acid sequence identity and have the identical structures: each monomer is formed by a core of two β-sheets surrounded by α-helices (*Nogales, Wolf & Downing, 1998*). The dynamic instability of MTs and their organization can be regulated by their interactions with microtubule associated proteins (MAPs) (*Struk & Dhonukshe, 2014*). The regulations of MT organization and dynamics by MAPs include MT polymerization and depolymerization (*Hamada, 2014*), nucleation (*Liu et al., 2014*; *Yi & Goshima, 2018*), stabilization (*Ambrose et al., 2011*; *Liu et al., 2013*), bundle (*Ebina, Ji & Sato, 2019*), cross-linking and severing (*Wang et al., 2018*), and connecting MTs to other cellular structures (*Chen et al., 2016*; *Perez et al., 1999*). Therefore, it is crucial to dissect the relationship between tubulin/MT and MAP. However, to date most plant expression vectors can only express one objective gene, so multiple vectors having to be constructed to study the interaction between tubulin/MT and MAP. To conduct a tobacco transient expression assay to study the interaction, we usually need to generate different expression vectors, and respectively transform them into *Agrobacterium tumefaciens*, and co-infiltrate inoculums into *Nicotiana benthamiana* leaves (*Drevensek et al., 2012*). In addition, because the objective genes are located on different vectors, the expression levels of genes are often imbalanced in one cell, and even only one gene is expressed in some cells (*Drevensek et al., 2012*). This imbalanced expression influences the accuracy of interaction between tubulin/ MT and MAP *in vivo*. So cells that express both genes have to be found to observe the colocalization and carry out a co-inmunoprecipitation (Co-IP) assay. Concerning the stable transformation of rice mediated by *Agrobacterium tumefaciens*, because of restrictions to plant expression vectors, the objective genes have to be respectively transformed into different calluses that are cultured to be transgenic plants. Hybridization has to be carried out to co-express the different objective genes in the same one plant (*Halpin et al., 1999*). To co-express tubulin and MAP *in vivo*, the different expression vectors need to be constructed and separately transformed into *Agrobacterium tumefaciens* both in tobacco transient expression and transgenic rice assays. Especially in transgenic rice assay, co-expressing objective genes in one plant through hybridization costs lots of time due to rice's long growth cycle. Therefore, constructing a co-expression vector can overcome the above troubles, simplifying the experiment procedures and greatly shortening the cycles of studies on protein interaction (*Halpin et al., 1999*).

To analyze the MT dynamic instability and subcellular localization of putative MAPs, live-cell imaging technology has been usually performed using fluorescent protein fusion to trace the MT dynamics and MAP localization (*Tian et al., 2015*; *Wang et al., 2018*). In addition, identifying positive transgenic plants by fluorescent proteins under a fluorescence microscope is more convenient and efficient than Western blot assay. The procedures of a Western blot assay are long period and complex operations, as it also needs to extract protein, run SDS-PAGE electrophoresis, transfer membrane, incubate antibody, conduct color development and other steps, as well as primary antibodies are expensive and sometimes unreliable (*Pillai-Kastoori, Schutz-Geschwender & Harford, 2020*).

Consequently, the two fluorescent proteins GFP and mCherry, of which excitation and emission light wavelengths are far away from each other, which can avoid the error caused by the overlap of the two fluorescent signals, were respectively fused with α-tubulin and MAP in the co-expression vector in this study.

A large variety of MAPs have been identified with a wide range of functions (*Struk & Dhonukshe, 2014*). Although lots of MAPs are function conserved in some species, it is still very meaningful to verify and characterize the MAPs in other species. So far, very little progress has been made in characterizing proteins which are associated with MTs in rice (*Deng et al., 2015*; *Guo et al., 2009*; *Yang et al., 2020*). It is necessary to verify the putative MAPs in rice and characterize their functions. Here, with the universal dual-transgene expression vector, we constructed a series of vectors that co-expressed α-tubulin and putative MAPs in rice, including GL7 (*LOC_Os07g41200*, GenBank accession: XM_015791791), OsKCBP (*LOC_Os04g57140*, GenBank accession: NM_001402453), OsCLASP (*LOC_Os04g42840*, GenBank accession: AK067306), and OsMOR1 (*LOC_Os01g60040*, GenBank accession: NM_001409114), and verified that they might be MAPs for the first time by tobacco transient expression assay. These results provide important support to further reveal their functions.

# METHODS AND RESULTS

## Plant materials

Rice cultivar Zhonghua 11 (*Oryza sativa* L. subsp. *japonica* cv Zhonghua11) plants were grown in paddy fields under natural conditions. *Nicotiana benthamiana* plants were grown in a greenhouse under 16 h of light, a diurnal temperature of 25 °C, and a nocturnal temperature of 20 °C.

## Strains and vectors

*Escherichia coli* strain: DH5α; *Agrobacterium tumefaciens* strain: EHA105. Plasmids: pCambia1301 (Fig. 1A; Vector S1, GenBank accession: AF234297), pCambia1391 (Vector S2, GenBank accession: AF234308), pBI221-GFP (Vector S3, GenBank accession: OR810744), pBI221-H2B-mCherry (Vector S4, GenBank accession: OR797710).

## Reagents and solutions

RNeasy Plant Mini Kit (cat. no. 74904; Qiagen, Hilden, Germany): extract total RNA from rice young inflorescence;

SuperScript®III First-Strand Synthesis System for RT-PCR (reverse transcription-polymerase chain reaction; cat. no. 18080-051; Invitrogen, Waltham, MA, USA): Synthesize first-strand cDNA from total RNA;

High-fidelity polymerase: KOD-Plus-Neo polymerase and mating reagents including 10×PCR buffer, 25 mM $MgSO_4$, and 2 mM dNTPs (cat. no. KOD-401; Toyobo, Osaka, Japan) were used in the PCR to amplify α-*tubulin* and *MAPs* ORFs from cDNA, and *GFP*, *mCherry*, *35S* promoter, and *NOS* terminators from relevant plasmids;

Taq DNA polymerase: 2×Taq MasterMix (Dye) (cat. no. CW0682; CWBio, Jiangsu, China) was used in the colony PCR;
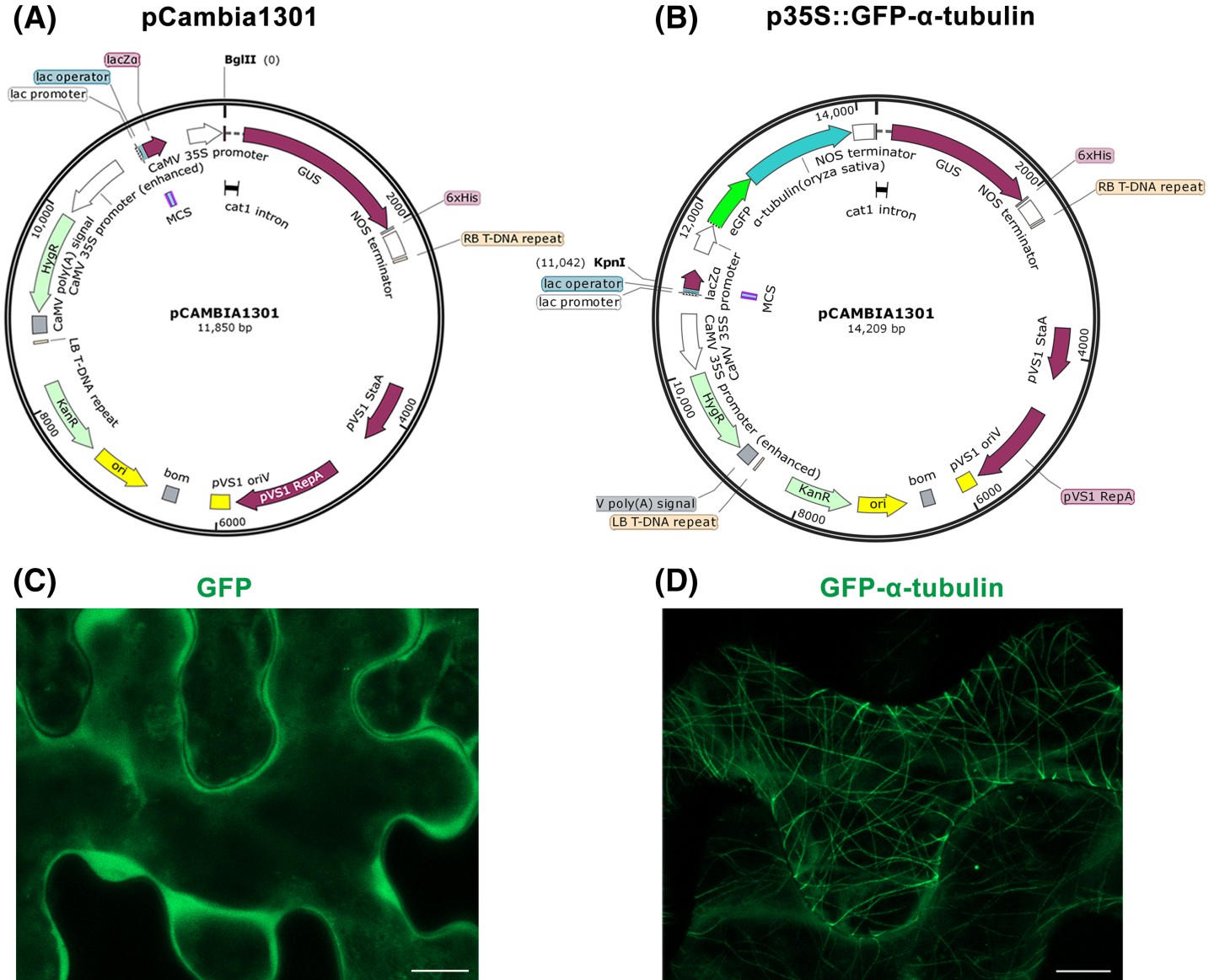

**Figure 1 Construction of p35S::GFP-α-tubulin vector and its expression in *Nicotiana benthamiana* leaf epidermal cells.** (A) Schematic of pCambia1301 vector. (B) Schematic of p35S::GFP-α-tubulin vector. (C) Subcellular localization of GFP protein. (D) Subcellular localization of GFP-α-tubulin as the p35S::GFP-α-tubulin vector is expressed. All micrographs are projections of confocal Z-stacks and bar = 10 μm.

Restriction endonuclease: BglII (cat. no. R0144; NEB, Ipswich, MA, USA) supplied with 10×NEBuffer 3.1 (cat. no. B7203; NEB, Ipswich, MA, USA); KpnI-HF (cat. no. R3142; NEB, Ipswich, MA, USA), AscI (cat. no. R0558; NEB, Ipswich, MA, USA), both supplied with 10×CutSmart buffer (cat. no. B7204; NEB, Ipswich, MA, USA);

Agarose (cat. no. e1714; BioWest, Bradenton, FL, USA): used for the analysis of nucleic acids by gel electrophoresis;

SYBR® Green I (cat. no. S7585; Sigma, Burlington, MA, USA): used for DNA stain in nucleic acid gel electrophoresis assay;

6×Gel loading dye (purple) (cat. no. B7025S; NEB, Ipswich, MA, USA): used for loading DNA samples on agarose during electrophoresis;

DNA marker: 1 kb DNA Ladder (cat. no. N3232S; NEB, Ipswich, MA, USA), and 100 bp DNA Ladder (cat. no. N3231S; NEB, Ipswich, MA, USA); used to mark the size of DNA fragments;

50×TAE (tris base + acetic acid + EDTA) buffer (cat. no. B49; Thermo Fisher Scientific, Waltham, MA, USA): a buffer used for nucleic acid gel electrophoresis; dilute it to 1× in water before use;

Monarch® DNA Gel Extraction Kit (cat. no. T1020; NEB, Ipswich, MA, USA): Purify DNA fragments and linear plasmids from an agarose gel;

Monarch® Plasmid Miniprep Kit (cat. no. T1010; NEB, Ipswich, MA, USA): Purify plasmids from *E. coli* culture medium;

NEBuilder® HiFi DNA Assembly Master Mix (cat. no. E2621; NEB, Ipswich, MA, USA): Assemble DNA fragments in seamless cloning assay;

LB (Luria-Bertani) medium: 1% tryptone (cat. no. LP0042B; Oxoid, Hampshire, England), 0.5% yeast extract (cat. no. LP0021; Oxoid, Hampshire, England), 1% NaCl (cat. no. S3014; Sigma, Burlington, MA, USA); add 1.5% agar (cat. no. 70101ES76; Yeasen, Shanghai, China) to obtain the solid medium; before usage it is sterilized at 121 °C for 20 min.

Antibiotic: dissolve kanamycin (cat. no. E004000; Sigma, Burlington, MA, USA) and rifampicin (cat. no. R3501; Sigma, Burlington, MA, USA) in water to obtain 50 mg/mL and 20 mg/mL stock solution, respectively; filtered with 0.22 μm membrane (cat. no. SLGP033RB; Millipore, Burlington, MA, USA) to remove bacteria;

Infiltration buffer: 10 mM MES (cat. no. M8250; Sigma, Burlington, MA, USA), pH 5.6, 10 mM MgCl$_2$ (cat. no. M8266; Sigma, Burlington, MA, USA), 150 μM Acetosyringone (cat. no. D134406; Sigma, Burlington, MA, USA);

MTSB buffer (microtubule-stabilizing buffer): 50 mM PIPES (cat. no. P1851; Sigma, Burlington, MA, USA), 5 mM MgSO$_4$ (cat. no. M7506; Sigma, Burlington, MA, USA), 5 mM EGTA (cat. no. E4378; Sigma, Burlington, MA, USA), pH 7.0, adjusted with KOH (cat. no. P5958; Sigma, Burlington, MA, USA).

### Equipment

PCR amplifier (Eppendorf Mastercycler pro);

Confocal microscopy (ZEISS 980 LSM);

Microwave oven (Galanz);

Apparatus for agarose gel electrophoresis (LiuYi DYY-6C);

UV (Ultraviolet) gel imager system (Cell Biosciences FluorChem FC2);

Ultramicro spectrophotometer (Thermo NANODROP ONE);

High-speed centrifuge (Eppendorf 5810 R);

High-pressure steam sterilization pot (FAEN D-1);

Ultra-low temperature freezer (Thermo Scientific);

Superclean bench (JD-VS-2S);

Constant temperature oscillator (PeiYing THZ-C-3).

TGreen blue light transmittance instrument (OSE-470L): observe and cut gels that contain nucleic acids;

Biochemical incubator (STIK B1-150A);

pH device (METTLER TOLEDO FE20);

Thermostatic water bath (Thermo, cat. no. TSGP05);

Electronic balance (Sartorius BSA323S);

Micropipettes (Gilson or Eppendorf);

Centrifuge tube and tips (Axygen): Various specifications;

Petri dish (Corning);

Mortar and pestle: 60 mm diameter;

Syringe: 1 mL specification;

Microscope slides (Leica, cat. no. 3800381);

Cover glasses (Leica, cat. no. 3800101): 0.17 mm thick; 22 × 22 mm;

## Procedures and results

### Construction of p35S::GFP-α-tubulin vector

**Step 1** Extracting total RNA

Weigh about 0.1 g of fresh rice young inflorescence, and grind to powder in liquid nitrogen; the next operations follow the RNeasy® Mini Handbook; mortar, pestle, tips, tubes, *etc.* are RNAase-free.

**Step 2** Reverse transcription

A total of 1 μg total RNA is used for reverse transcription to obtain cDNA by SuperScript®III First-Strand Synthesis System for RT-PCR; the operations follow the instructions of the Kit; temperature is controlled with a PCR amplifier.

**Step 3** Amplification of *GFP*, α-*tubulin* ORF and *NOS* terminator sequences, and vector linearization

The plasmid pBI221-GFP, cDNA in **Step 2**, and plasmid pCambia1391 are used as templates to respectively amplify *GFP*, α-*tubulin* ORF and *NOS* terminator sequences; sequence-specific primers are listed in Table S1; high-fidelity polymerase KOD-Plus-Neo is used in the PCR, and reaction setup and thermocycling conditions for PCR are performed according to the protocol for KOD-Plus-Neo polymerase. pCambia1301 is digested with restricted enzyme BglII to linearization, and the reaction system and operations follow the product instructions.

**Step 4** Electrophoresis and gel extraction of target fragments

• Agarose powder is mixed with electrophoresis buffer (1×TAE buffer) to 1% concentration (w/v), and heated in a microwave oven until completely melted; DNA stain dye (SYBR® Green I) is added to the gel (final concentration 0.5 μg/mL); cool the solution to 60 °C and pour into a casting tray containing a sample comb and allow to solidify at room temperature.

• Gel is placed into the electrophoresis apparatus and immersed in 1×TAE buffer; DNA fragments in **Step 3** mixed with loading buffer, and DNA marker are pipetted into the sample wells.

• Electrophoresis is performed immediately at a voltage between 60–100 V; when adequacy has occurred, stop the electrophoresis.

• Transfer the gel to a UV gel imager system for imaging, and excise under the TGreen blue light transmittance instrument; the excised gel containing DNA fragments should be as small as possible, and exposure time under the UV light should be kept as short as possible to minimize the damage to the DNA.

• Use Monarch® DNA Gel Extraction Kit to extract the DNA fragments from agarose gels; operations follow the protocol of kit.

• Measure DNA fragment concentrations with the ultramicro spectrophotometer; wash the sample chamber three times with deionized $H_2O$ before use. Immediately use extracted DNA fragments for seamless cloning or store them at −20 °C.

**Step 5** Seamless cloning (Gibson assembly)

The extracted DNA fragments (including *GFP*, *α-tubulin*, *NOS* terminator, and linearized pCambia1301), deionized $H_2O$, and Assembly Master Mix are mixed in proportion according to the concentrations on ice; the mixture is incubated at 50 °C and transformed to DH5α competent *E. coli*; use kanamycin (final concentration 50 μg/mL) LB plate to select positive bacterial colonies; all the operations follow the NEBuilder® HiFi DNA Assembly Master Mix protocol.

**Step 6** *E. coli* colony PCR, sequencing, and plasmid purification

• Prepare a PCR amplification reaction mixture; Taq DNA polymerase is used for colony PCR; PCR amplification reaction setup is conducted following the product instruction; amplification primers are listed in Table S1.

• Stab a transformed colony with a sterile toothpick and swirl cells from the colony into the amplification reaction mixture in a superclean bench; gently mix each reaction.

• Perform PCR and analyze the PCR products using standard 1% (w/v) agarose gel electrophoresis; thermocycling conditions is conducted following the product instructions.

• Inoculate 0.5 mL of the kanamycin (final concentration 50 μg/mL) LB medium into a 2 mL sterile tube with an isolated colony that is confirmed by colony PCR and incubate at 37 °C at 200 rpm until the culture becomes turbid.

• Transfer 0.2 mL inoculum into a new sterile tube to send to the company (Tsingke Biotech Co., Ltd., Beijing, China) for sequencing verification; add sterile 75% glycerin to the remaining inoculum to a final concentration of 15% and store at −80 °C.

• Inoculate 4 mL of the kanamycin (final concentration 50 μg/mL) LB medium into a sterile vessel that allows some aeration (culture tubes on a roller drum) with a stored inoculum that had been verified by sequencing; incubate at 37 °C at 200 rpm about 12–16 h.

• Pellet bacterial culture by centrifugation and purify plasmid by Monarch® Plasmid Miniprep Kit; the operations follow the Kit protocol; plasmid DNA is eluted with nuclease-free water (pH 7–8.5) and stored at −20 °C.

**Step 7** Plasmid transformation into *Agrobacterium tumefaciens* competent cells and colony PCR

The plasmid p35S::GFP-α-tubulin (Fig. 1B; Vector S5, GenBank accession: OR797711) purified in **Step 6** is transformed into *Agrobacterium tumefaciens* EHA105 competent

cells, following the chemically competent cells transformation protocol; use kanamycin (final concentration 50 μg/mL) and rifampicin (final concentration 20 μg/mL) LB plate to select positive bacterial colonies and incubate at 28 °C. After 2–3 d, carry out colony PCR following the **Step 6** operations to further verify the positive colonies.

**Step 8** Tobacco transient expression assay

• Grow *N. benthamiana* to the age of 4 weeks; water the *N. benthamiana* plants 1 d (day) before the infiltration to make the infiltration easier.

• Separately incubate *Agrobacterium* cultures p35S::GFP-α-tubulin, and P19 (*Garabagi et al., 2012*), a viral suppressor of gene silencing, overnight at 28 °C at 200 rpm in 3 mL LB medium with kanamycin (final concentration 50 μg/mL) and rifampicin (final concentration 20 μg/mL).

• Inoculate *Agrobacterium* from the LB medium into the fresh medium for enlarged cultivation; 1/100 dilution for overnight culture or 1/25 for about 8 h culture.

• Incubate at 28 °C at 200 rpm for about 8–12 h, so that the *Agrobacterium* cell is in the logarithmic growth phase; the best OD600 (optical density at 600 nm) value should be 0.5–0.8.

• Centrifuge *Agrobacterium* culture in 50 mL Corning tube for 15 min at 5,000×*g* at room temperature to pellet the cells.

• Prepare infiltration solution, and resuspend *Agrobacterium* pellet in 10 mL of infiltration solution to a desired OD600. **Note**: Mix different kinds of *Agrobacteria*, with the desired OD600 (usually 0.1–0.5) for each one; use a larger OD600 value for *Agrobacteria* expressing lower protein while using a smaller OD600 value for *Agrobacteria* expressing abundant protein in *N. benthamiana*.

• Let the inoculum stand and away from light for 2–3 h.

• Choose the healthiest looking plants and leaves; leaves 3–5 from top to bottom are usually chosen.

• Swirl the inoculum in the tube, and use a 1 mL blunt syringe to gently inject the inoculum into the underside of the leaf. Successful infiltration is often observed as a spreading 'wetting' area in the leaf.

• Place the infiltrated plants in the dark for 1 d, then grow them under 16 h day/8 h night at 22 °C.

• After 2–5 d of infiltration, cut the infiltrated leaf to an appropriate size, and immediately immerse it in MTSB buffer on a microscope slide, and carefully cover it with cover glass.

• Immediately transfer the microscope slide into the sample chamber of confocal microscopy to detect the fluorescence. Fluorescence is recorded sequentially after an excitation at 488 nm for the GFP and at 561 nm for the mCherry; the micrograph is projections of Z-stack confocal images (*Drevensek et al., 2012*).

Through above **Steps 1–6,** the *GFP*, α-*tubulin* ORF, and *NOS* terminator fragments have been fused and inserted into the pCambia1301 to construct p35S::GFP-α-tubulin vector, in which the α-*tubulin* gene is driven by *CaMV 35S* promoter, a strong promoter is inherent in the pCambia1301 itself and drive *GUS* gene expression before.

Furthermore, the p35S::GFP vector previously constructed by our laboratory to express GFP protein was used as a control. With the tobacco transient expression assay, the GFP protein distributed into the cytoplasm (Fig. 1C) but the GFP-α-tubulin protein localized to MTs (Fig. 1D), which suggested that the rice α-tubulin was able to localize to MTs.

By the tobacco transient expression assay (**Step 8**), it has been verified that α-tubulin was successfully expressed and localized to MTs (Fig. 1D). So the p35S::GFP-α-tubulin vector can be used for the subsequent procedures.

### Construction of p35S::GFP-α-tubulin-p35S::mCherry vector

**Step 9** Amplification of *35S*, *mCherry*, and *NOS* terminator sequences, and vector linearization

The *35S* promoter is amplified by two rounds of PCR. The plasmid pBI221-H2B-mCherry is used as a template to amplify with the first primer pair; the PCR product is also used as a template to amplify with the second primer pair. Through the two rounds of PCR, we obtain a *35S* promoter sequence with the restriction enzyme cutting sites XbaI, KpnI/Acc65I at its 3′ terminus. Using plasmid pBI221-H2B-mCherry as a template, the *mCherry* sequence is amplified with a primer pair, and the AscI restriction site is added to the 3′ terminus of the *mCherry* sequence. By adding restriction endonuclease sites XbaI, KpnI/Acc65I at the 5′ terminus and AscI at the 3′ terminus of *mCherry*, another gene, *e.g.*, *MAP* could be fused with *mCherry* at its 5′ or 3′ terminus, facilitating the universality of vector's application. Plasmid pBI221-H2B-mCherry is also used as a template to amplify the *NOS* terminator with a primer pair. All sequence specific primers are listed in Table S1. p35S::GFP-α-tubulin vector is digested with restricted enzyme KpnI-HF to linearization. All operations are the same as in **Step 3**.

**Step 10** Electrophoresis and gel extraction of target fragments

The PCR products of *35S*, *mCherry* and *NOS* terminator, and linearized p35S::GFP-α-tubulin vector are subjected to electrophoresis and gel extraction. All operations are the same as in **Step 4**.

**Step 11** Seamless cloning (Gibson assembly).

The extracted DNA fragments *35S* promoter, *mCherry*, and *NOS* terminator are fused and inserted into the p35S::GFP-α-tubulin vector at the KpnI site to construct p35S::GFP-α-tubulin-p35S::mCherry vector (Fig. 2A; Vector S6, GenBank accession: OR797712). All operations are the same as in **Step 5**.

**Step 12** *E. coli* colony PCR, sequencing, and plasmid purification

Through the colony PCR, sequencing, and plasmid purification, we obtain the p35S::GFP-α-tubulin-p35S::mCherry vector. All operations are the same as in **Step 6**.

**Step 13** Plasmid transformation into *Agrobacterium tumefaciens* competent cells and colony PCR.

Transform the p35S::GFP-α-tubulin-p35S::mCherry vector into *Agrobacterium tumefaciens* EHA105 competent cells and confirm it by colony PCR. All operations are the same as in **Step 7**.
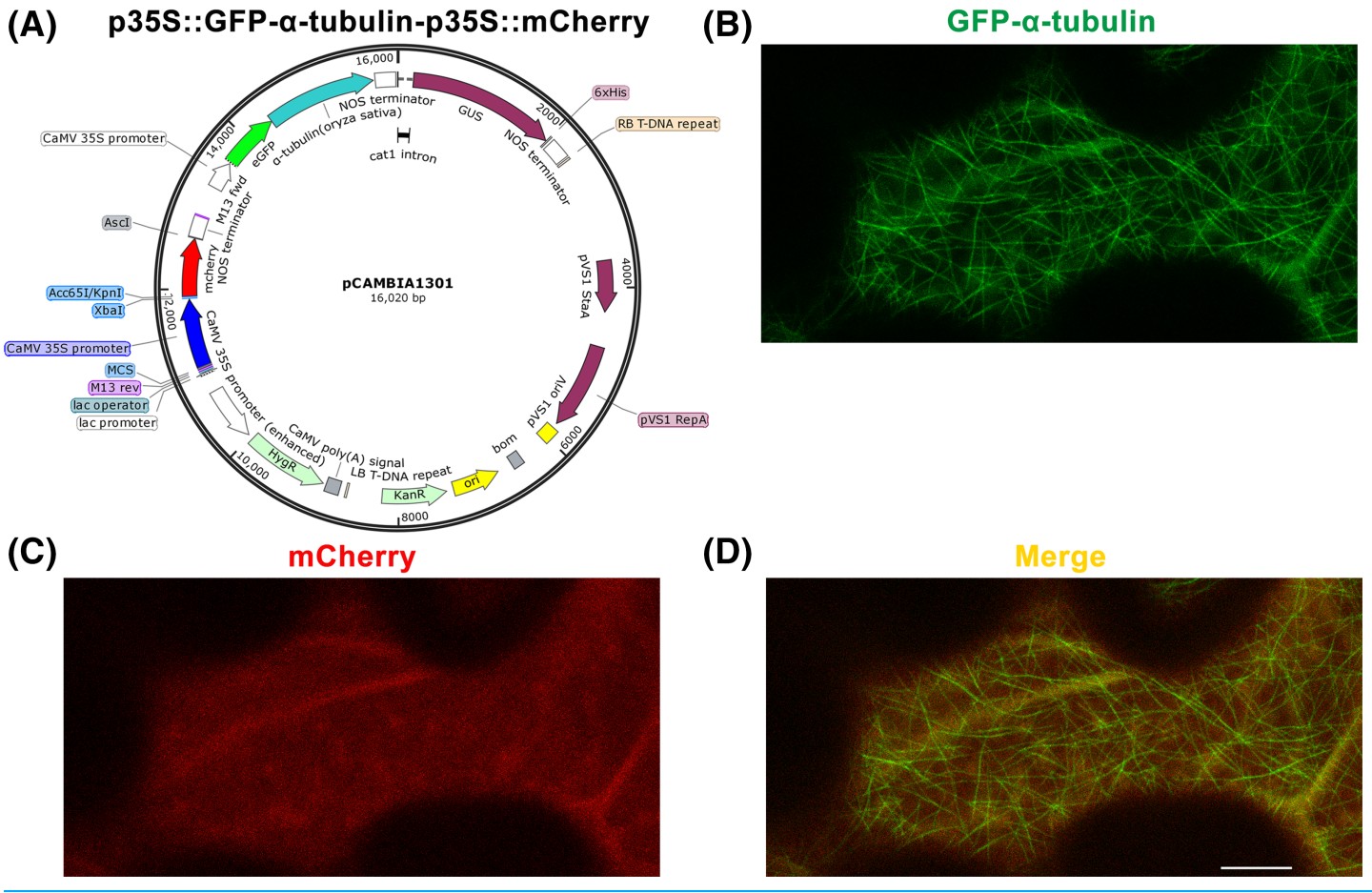

**Figure 2 Construction of p35S::GFP-α-tubulin-p35S::mCherry vector and its expression in *Nicotiana benthamiana* leaf epidermal cells.** (A) Schematic of p35S::GFP-α-tubulin-p35S::mCherry vector. (B) Subcellular localization of GFP-α-tubulin as the vector is expressed. (C) Subcellular localization of mCherry as the vector is expressed. (D) Micrograph of GFP-α-tubulin merged with mCherry. All micrographs are projections of confocal Z-stacks and bar = 10 μm.

**Step 14** Tobacco transient expression assay.

Through this assay, GFP-α-tubulin and mCherry are transiently co-expressed in *N. benthamiana* leaves and detected by confocal microscopy. All operations are the same as in **Step 8**.

Based on the p35S::GFP-α-tubulin vector, we constructed a universal dual-transgene expression vector p35S::GFP-α-tubulin-p35S::mCherry (Fig. 2A) through above **Steps 9–12**, in which *GFP-α-tubulin* and *mCherry* genes are driven by *35S* promoters, respectively, and could be simultaneously expressed. Through **Steps 13–14**, the vector p35S::GFP-α-tubulin-p35S::mCherry was transformed into *N. benthamiana* leaves, and GFP-α-tubulin and mCherry were co-expressed in leaf epidermal cells (Figs. 2B–2D). The availability of the p35S::GFP-α-tubulin-p35S::mCherry vector was validated by the

observation that GFP-α-tubulin localized to MTs (Fig. 2B) and mCherry distributed into the cytoplasm (Fig. 2C).

### Validation of p35S::GFP-α-tubulin-p35S::mCherry vector

To further verify the availability and universality of the p35S::GFP-α-tubulin-p35S:: mCherry vector, several putative *MAP* genes in rice including *GL7* (Sequence S1), *OsKCBP* (Sequence S2), and *OsCLASP* (Sequence S3) were inserted into the 3′ terminus of *mCherry* in the vector respectively, and *OsMOR1* (Sequence S4) was inserted into the 5′ terminus of *mCherry*. All of the constructed dual-transgene vectors were transiently expressed in *N. benthamiana* leaves, respectively.

*GL7*, a major quantitative trait locus (QTL) for grain size diversity in rice, encodes a protein homologous to *Arabidopsis thaliana* LONGIFOLIA proteins (*Wang et al., 2015*). The GL7 protein has 19–22% amino acid sequence identity with the LONGIFOLIA1 and LONGIFOLIA2 proteins (*Wang et al., 2015*; Fig. S1). LONGIFOLIA2 is a MAP which may be involved in directing cell expansion by recruiting TON1 to cortical MT arrays (*Drevensek et al., 2012*; *Lee et al., 2006*). So the GL7 appears to be a MAP in rice, but the direct experimental evidence is lacking. Here, we used the p35S::GFP-α-tubulin-p35S:: mCherry vector to construct the p35S::GFP-α-tubulin-p35S::mCherry-GL7 vector and transformed it into *N. benthamiana* leaves. The results suggested that mCherry-GL7 colocalized with GFP-α-tubulin and presented a punctate pattern along MTs (Fig. 3A), consistent with the subcellular localization of LONGIFOLIA2/TRM1 in *N. benthamiana* leaf epidermal cells (*Drevensek et al., 2012*). This result supplies the experimental evidence that GL7 might be a MAP.

Kinesin-like calmodulin-binding protein (KCBP), a calmodulin-binding kinesin of the Kinesin-14 sub-family (*Richardson, Simmons & Reddy, 2006*), occurs solely within the plant kingdom and acts as a hub that brings together MTs and actin filaments to modulate the cytoskeleton during trichome formation in *Arabidopsis thaliana* (*Tian et al., 2015*). The homologous protein of KCBP in rice has 70% amino acid sequence identity with KCBP (Fig. S2) but has not been studied. By the p35S::GFP-α-tubulin-p35S::mCherry vector, we constructed p35S::GFP-α-tubulin-p35S::mCherry-OsKCBP vector and transformed it into *N. benthamiana* leaves. The results showed that mCherry-OsKCBP colocalized with GFP-α-tubulin and somewhat decorated MTs as a punctate pattern in *N. benthamiana* leaf epidermal cells (Fig. 3B), indicating that the OsKCBP might be a MAP.

Cytoplasmic linker associated proteins (CLASPs) are considered as MT plus end-tracking proteins and well-studied MAPs found across plant, fungus, and animal systems. CLASPs have a broad range of functions in cell motility, cell expansion and proliferation, and localize at kinetochores, the mitotic spindle midzone, centrosomes, the Golgi network, and the cell cortex (*Bratman & Chang, 2008*; *Efimov et al., 2007*; *Kumar et al., 2009*; *Mimori-Kiyosue et al., 2005*; *Ruan et al., 2018*). The functions of CLASPs have been characterized for stabilizing overlapping MTs in mitotic spindles of fission yeast (*Bratman & Chang, 2007*), promoting MT bundling in metaphase spindle in fission yeast

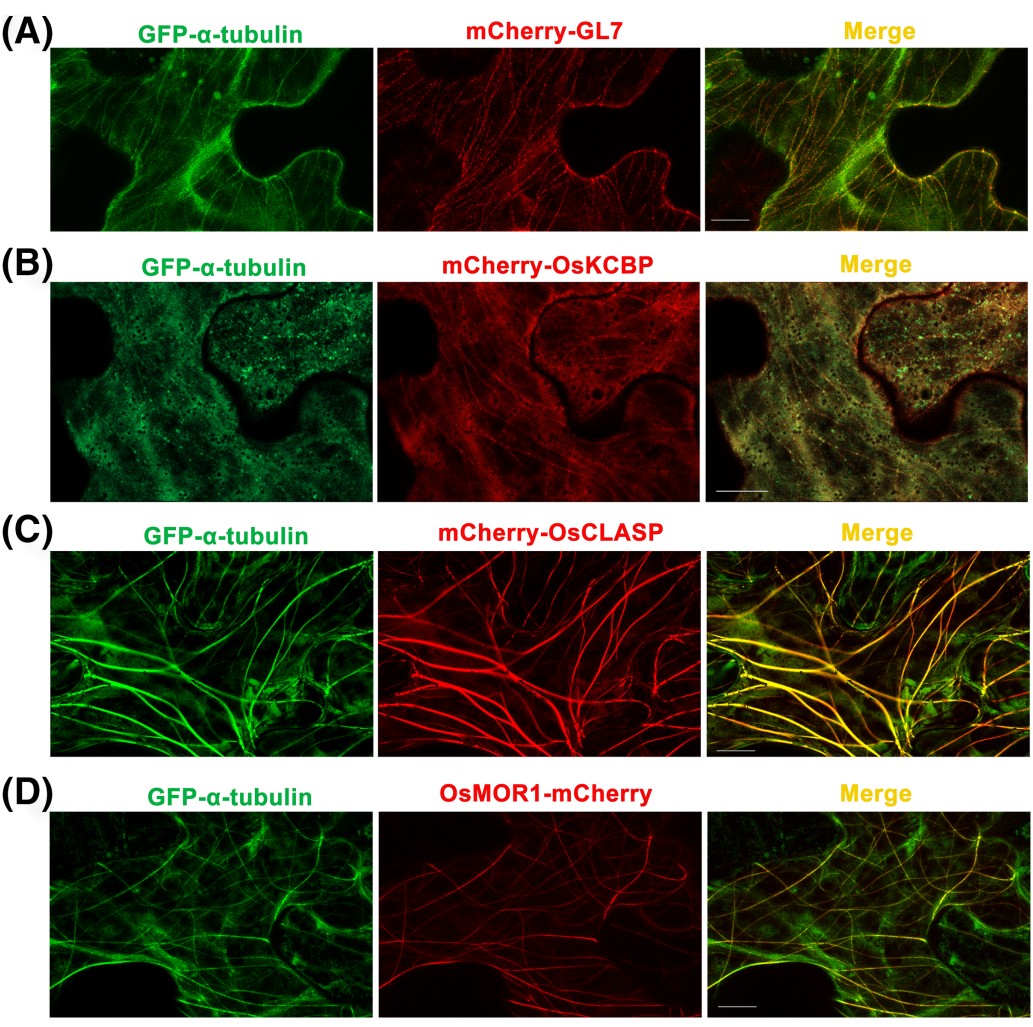

**Figure 3 Co-expressions of rice α-tubulin and MAPs as the dual-transgene vectors are transformed into *Nicotiana benthamiana* leaf epidermal cells respectively.** (A) Co-expressions of rice GFP-α-tubulin and mCherry-GL7. (B) Co-expressions of rice GFP-α-tubulin and mCherry-OsKCBP. (C) Co-expressions of rice GFP-α-tubulin and mCherry-OsCLASP. (D) Co-expressions of rice GFP-α-tubulin and OsMOR1-mCherry. All micrographs are projections of confocal Z-stacks and bar = 10 μm.

(*Ebina, Ji & Sato, 2019*), inducing MT pausing in *Drosophila* S2 cells (*Sousa et al., 2007*), stimulating MT rescues at leading cell edges (*Mimori-Kiyosue et al., 2005*), suppressing MT catastrophes (*Aher et al., 2018*; *Al-Bassam et al., 2010*), and promoting γ-tubulin-dependent MT nucleation at the Golgi (*Efimov et al., 2007*). CLASP localizes to MT lattice and induces MT bundles at transient high expression level in tobacco leaf epidermal cells, but enriches at growing MT plus ends at stable low expression in *Arabidopsis thaliana* leaf guard cells (*Ambrose et al., 2007*). CLASP is a very important MAP, but it has not been studied in rice. The OsCLASP has 63% amino acid sequence identity to AtCLASP (Fig. S3). In our study, the OsCLASP coding sequence was cloned into p35S::GFP-α-tubulin-p35S::mCherry vector and co-expressed with α-tubulin in *N. benthamiana* leaf epidermal cells. The results showed that mCherry-OsCLASP colocalized with GFP-α-tubulin to MT

lattices and induced MT bundles (Fig. 3C), which is similar with AtCLASP location at transient high expression level in tobacco leaves. The OsCLASP always kept great fluorescence to MTs and merely distributed into the cytoplasm (Fig. 3C), suggesting its strong bonding ability to MTs. This implies that OsCLASP might be a MAP and have similar functions with its homologs in other species.

Microtubule organization 1 (MOR1) of *Arabidopsis thaliana* belongs to the MAP215 family of MAPs (*Whittington et al., 2001*). MOR1 regulates cortical MT organization and function, as well as the structure and function of MT arrays during mitosis and cytokinesis in *Arabidopsis* root (*Chen et al., 2022*; *Kawamura et al., 2006*). MAP215 proteins are essential MT polymerases that use multiple TOG domains to bind MT plus ends and catalyze fast MTs (*Ayaz et al., 2014*). These proteins may function as MT polymerases by adding new tubulin dimers to the plus ends, and localize to MT plus ends, MT organizing centers, kinetochores, and MT lattices. Here, we constructed the p35S::GFP-α-tubulin-p35S::OsMOR1-Cherry vector and transformed it into *N. benthamiana* leaves. The transient expression showed that OsMOR1-Cherry colocalized with GFP-α-tubulin to MTs, and presented a linear pattern along MTs, which suggested an association with cortical MTs (*Thawani, Kadzik & Petry, 2018*; Fig. 3D). In addition, the OsMOR1 has 62–68% amino acid sequence identity with the *Arabidopsis* MOR1 (Fig. S4). Thus, we speculate that OsMOR1 might be a MAP.

By the above tobacco transient expression assays, we validated the availability and universality of the dual-transgene expression vector (p35S::GFP-α-tubulin-p35S::mCherry), and provided experimental evidence that GL7, OsKCBP, OsCLASP, and OsMOR1 might be MAPs for the first time.

## DISCUSSION

The locations of objective genes on different vectors often lead to imbalanced expression levels, even alternative expression in a cell. To accurately reflect the interaction between proteins *in vivo* in transient expression assay, especially when comparing the strength of protein interaction between wild-type protein and point mutation protein with other interacting proteins, the expressions of genes must be simultaneous and equivalent in a cell. One of the strategies employed to construct multicistronic vectors is the insertion of an internal ribosomal entry site (IRES) between genes, but IRES also has a major limitation that the translation efficiency of a gene placed after the IRES is much lower than that of a gene located before IRES (*Kim et al., 2011*). The self-cleaving 2A peptides have been usually used to construct multicistronic vectors (*Kim et al., 2011*; *Ralley et al., 2004*; *Yu et al., 2016*), because of its small size and high cleavage efficiency between genes upstream and downstream of 2A peptide. Despite the advantages of the 2A peptides, there are still some limitations, including changes in cleavage efficiency among various 2A peptides and in different contexts, and 2A peptide fusion at the C terminus of upstream protein. The self-cleaving 2A peptide could be a good candidate for constructing multicistronic vectors in rice, and a few studies focused on the expression of multiple genes linked with 2A sequences in rice endosperm (*Ha et al., 2010*; *Jeong et al., 2021*). Therefore, it is necessary

to further understand cleavage characteristics and the factors affecting the cleavage in rice context.

Seamless cloning technology based on Gibson assembly has lots of advantages than the traditional double enzyme digestion method to construct vectors. The double enzyme digestion method is limited by sites of restriction enzymes, as well as the golden gate assembly requires specialized restriction enzymes. Gibson assembly promotes the efficiency of constructing vectors, especially for the recombination of multiple fragments (*Gibson, 2011*). This construction is not restricted by restriction sites and achieves seamless cloning by adding or subtracting bases by overlapping sequences in primers. Here, we used the Gibson assembly to construct a dual-transgene expression vector (p35S::GFP-α-tubulin-p35S::mCherry). Based on this vector, we subsequently constructed four vectors that would respectively co-express rice α-tubulin and putative MAP. The *GL7*, *OsKCBP*, and *OsCLASP* were inserted into the 3′ terminus of *mCherry* in the vector respectively, and *OsMOR1* was inserted into the 5′ terminus of *mCherry*. In the tobacco transient expression assay, α-tubulin and putative MAP expressions were simultaneous and exhibited comparable expression levels in cells (Fig. 3), although α-*tubulin* and *MAP* were driven by respective *35S* promoters. All of the four putative MAPs were successfully co-expressed with α-tubulin and localized to MTs (Fig. 3), suggesting that the dual-transgene expression vector was available and universal. Consequently, we suggest that the dual-transgene expression vector (p35S::GFP-α-tubulin-p35S::mCherry) can be applied for MT dynamic trace, MAP verification, colocalization observation, Co-IP assay, and stable transformation of rice. Given tubulin's conservation, the application of the dual-transgene expression vector (p35S::GFP-α-tubulin-p35S::mCherry) may be expanded to other plants.

In this study, we provided the experimental evidence for the first time that GL7, OsKCBP, OsCLASP, and OsMOR1 might be MAPs with the p35S::GFP-α-tubulin-p35S::mCherry vector in the tobacco transient expression assays. However, it is not definite to identify MAPs only by colocalization in the tobacco transient expression assay. As the expressions are ectopic, transient and overexpressed, and the existence of indirect interactions, it may not accurately reflect the behavior of these proteins in their native context. To identify MAPs in rice, it is necessary to conduct other validation experiments, including MT coprecipitation assay *in vitro* and observation of colocalization with MTs *in vivo* in rice, and so on. Moreover, MT colocalization observation in tobacco transient expression assay is confused by many factors (*Guo et al., 2023*). Such as the properties of MAPs determine the specificity, strength and dynamic of the bond to MTs, and the high fluorescence background ascribed to the transient overexpression which produces excess protein causes it hard to observe colocalization (*Drevensek et al., 2012*; *Guo et al., 2023*). In our study, the OsCLASP and OsMOR1 proteins localized to MTs, but the GL7 and OsKCBP proteins weakly localized to MTs (Fig. 3). This difference may be due to the protein's properties and the high fluorescence background. Overall, by the dual-transgene expression vector and tobacco transient expression assay, we initially consider that GL7, OsKCBP, OsCLASP, and OsMOR1 are MAPs in rice, but the confirmation needs extra validation experiments.

## CONCLUSIONS

In this study, we constructed a universal dual-transgene expression vector (p35S::GFP-α-tubulin-p35S::mCherry) based on the plant expression vector pCambia1301. The p35S::GFP-α-tubulin-p35S::mCherry vector can co-express rice α-tubulin and MAP which were respectively fused with fluorescent proteins GFP and mCherry. At last, we validated the availability and universality of the p35S::GFP-α-tubulin-p35S::mCherry vector and verified that GL7, OsKCBP, OsCLASP, and OsMOR1 might be MAPs for the first time with the vector by tobacco transient expression assays. Application of the vector would facilitate studies on interactions between tubulin/MT and MAP in transient expression systems or transgenic rice.

## ACKNOWLEDGEMENTS

We thank Jingquan Li at the Institute of Botany, Chinese Academy of Sciences (IB, CAS) for her contributions to confocal microscopy.

### Funding

This work was supported by the National Spark Project of China (No. 2015GA740088) and the Undergraduate Innovation and Entrepreneurship Project of Dezhou University of China (No. X202210448037). The funders had no role in study design, data collection and analysis, decision to publish, or preparation of the manuscript.

### Grant Disclosures

The following grant information was disclosed by the authors:
National Spark Project of China: 2015GA740088.
Dezhou University of China: X202210448037.

### Competing Interests

The authors declare that they have no competing interests.

### Author Contributions

- Chenshan Xu conceived and designed the experiments, performed the experiments, analyzed the data, prepared figures and/or tables, authored or reviewed drafts of the article, manage tobacco materials, and approved the final draft.
- Xiaoli Zhu performed the experiments, authored or reviewed drafts of the article, and approved the final draft.
- Aihong Xu performed the experiments, authored or reviewed drafts of the article, and approved the final draft.
- Jian Song analyzed the data, prepared figures and/or tables, authored or reviewed drafts of the article, and approved the final draft.
- Shuxia Liang analyzed the data, prepared figures and/or tables, and approved the final draft.

## DNA Deposition

The following information was supplied regarding the deposition of DNA sequences:

The sequence data are available in National Center for Biotechnology Information: XM_015791791, NM_001402453, AK067306, NM_001409114, AF234297, AF234308, OR810744, OR797710, OR797711, OR797712.

## Data Availability

The raw images are available in the Supplemental Files. The raw data shows all maps and sequences of related vectors, and micrographs.

## Supplemental Information

Supplemental information for this article can be found online at http://dx.doi.org/10.7717/peerj.18118#supplemental-information.

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
