# Peer review of "Construction and validation of co-expression vector for rice alpha tubulin and microtubule associated protein respectively fused with fluorescent proteins"

_PeerJ, doi:10.7717/peerj.18118_

## Round 0.1 · original submission · Minor Revisions

· Academic Editor

Minor Revisions

The manuscript was reviewed by three independent experts in the field. While all reviewers found the work interesting, reviewers 2 and 3 raised several valid concerns. They provided detailed comments, highlighting areas where the manuscript needs improvement.

·

Basic reporting

No comment

Experimental design

No comment

Validity of the findings

No comment

Additional comments

• Line 295: It should be “ On the basis of…”
• Line 324: space in actinfilaments
• Line 637: Did you mean altered expression in a cell?
• Line 398: What do you mean by tubulin’s conservation?

Reviewer 2 ·

Basic reporting

The study aims to verify the potential and universality of the GFP-α tubulin-mCherry vector and its application in identifying microtubule-associated proteins (MAPs) in rice. The authors inserted three putative MAP genes of rice under 35S promoter into the vector and conducted transient expression assays in N. benthamiana leaves to assess colocalization and subcellular localization patterns with references to microtubules, providing evidence that the identified proteins were potential MAPs. This vector system is a convenient and fast method for studying plant MAPs. It provides valuable insights into potential MAP candidates, although it is not a definite/confirmative method as it has several limitations that have not been discussed in the MS.

Experimental design

No Comment

Validity of the findings

1, Line#323-324 -"This result supplies the experimental evidence that GL7 is a MAP". It's important to note that transient expression may not accurately reflect the behavior of these proteins in their native context, and results obtained from such systems should be interpreted cautiously. This claim is based on a colocalization experiment, which is a valuable but not definitive method; it also needs in vitro assays to support rice GL7 as a MAP. I suggest that the authors modify this statement to reflect the limitations of the colocalization experiment.
2, Line#332-333: What is the percentage identity of rice KCBP with Arabidopsis KCBP? Again, a single co-localization experiment is not sufficient to label the protein as MAP. Authors must take care when writing such a statement.

Additional comments

1, Line#164/line#270- spelling correction "construction"
2, Line#270- "GFP-α tubulin-mCherry vector" This is a confusing way of writing the duel gene construct, as it suggests that everything is fused and controlled by a single promoter, which is not the case.
3, Line#328-"actinfilaments" to "actin filaments"
4, Regarding Figure 3b, I will encourage the authors to replace this image with a better one. The current image does not show great MT fluorescence with GFP Puncta.

Reviewer 3 ·

Basic reporting

The authors constructed a co-expression vector for rice alpha-tubulin and microtubule-associated proteins fused to fluorescent proteins GFP and mCherry. The application of the co-expression vector constructed might facilitate studies on the interaction between tubulin/MT and MAP in tobacco transient expression systems or transgenic rice. They constructed a universal dual-transgene expression vector (GFP- alpha tubulin-mCherry) based on the plant expression vector pCambia1301 using Gibson assembly. The GFP-alpha tubulin-mCherry vector was shown to co-express rice alpha-tubulin and MAPs which were respectively fused with GFP and mCherry. They validated the GFP-alpha tubulin-mCherry vector, and verified that GL7, OsKCBP, OsCLASP and OsMOR1 were MAPs for the first time with the vector by tobacco transient expression assays.

There are some typos and grammatical errors throughout the manuscript. References are requested in many areas throughout the manuscript. Raw data was provided as image files but the field of image looks blurred and misses scale bars in many places. Validation results for the use of vector is slightly acceptable but the claims made might require more validation (refer to the comments attached). Abstract should be seriously revised as it fails to provide context on the different MAPs used and therefore makes the title unsuitable for the manuscript.

Experimental design

The paper is a methods paper and gives details on all instruments and reagents used with an exception of real few. Research fills a simple gap using common Gibson assembly. Methods are described in sufficient detail & information to replicate. However, the manuscript lacks control experiments and images and claims that are not backed using literature.

Validity of the findings

Underlying data is provided as images and validity of the cloning is checked using four different MAPs. However, the discussion and conclusions section is too brief.

Additional comments

The title containing the term ‘Application’ is far-fetched since the authors are merely validating their vector and not applying anything.
Lines 16 and 38: Write what alpha and beta tubulins exactly is
Lines 41-43: more references for each claim
Line 49-72: References missing for each claim
Line 79: More recent reference required
Line 96: Typo in Agrobacterium tumefaciens throughout the manuscript
Line 104: Typo on Total
Line 130: Typo in Rifampicin
Line 217: Which company?
Line 242: OD abbreviations? Write the wavelength in nanometers.
Line 251: What are the size of the leaves? Photos of the leaves cut will be beneficial.
Line 259: Where is the micrograph reference
Line 262: construct typo
Line 266: Figure 1C has some background at the bottom and bright spots. What could the spots potentially be?
Line 293: Through instead of though
Lines 298-300 could be moved after line 272
Line 307: Sequences for GL7, OsKCBP, OsCLASP, OsMOR1
Line 313: Show the sequence similarity (run a BLAST or show sequence comparison) at least in the supplementary section
Line 325: Space between actin and filaments
Line 329: Figure 3b shows punctate dots. What do they represent? Also, why is the mCherry image different between each of those images – a, b, c, and d
Line 357: Why is OsMOR1 cloned before mCherry whereas GL7, OsKCBP and OsCLASP is cloned after mCherry
Line 362: Why is OsCLASP and OsMOR1 more localized with MT in figures 3c and 3d? Whereas, GL7 and OsKCBP has mCherry suspended through the cell or not as localized throughout MT?
Line 370: One of the strategies…
Line 404: MAPs typo
Line 413: Funding details missing

Where are the controls that express only mCherry or GFP-tubulin or no gene-carrying control?
What is the source of GFP and mCherry?
Figure 1 a and b: Plasmid labels are too small to see
Figure 2: Scale bars missing in 2 b and 2 c
Figure 3: Scale bars missing in the first two images of figure 3a-d

---

## Round 0.2 · Minor Revisions

· Academic Editor

Minor Revisions

Thank you for revising the manuscript. The current version is significantly improved; however, it still requires minor revisions to correct the English language and typographical errors. Reviewer 1 noted that the manuscript contains some grammatical mistakes. The authors should carefully review the manuscript to rectify these errors. Additionally, I suggest consistently writing 'α-tubulin' and 'β-tubulin' instead of 'α tubulin' and 'β tubulin' throughout the manuscript. Similarly, enzyme names, such as 'Kpn I,' should be written as 'KpnI.' Please address these and any similar errors found in the manuscript.

Reviewer 2 ·

Basic reporting

No Comments

Experimental design

No comments

Validity of the findings

No comments

Additional comments

In the revised manuscript, the authors have diligently addressed the reviewers' suggestions and made appropriate modifications to minor and major issues raised during the review process. These revisions have significantly enhanced the manuscript's quality. However, there are still a few grammatical errors that need to be corrected.

Reviewer 3 ·

Basic reporting

No comment

Experimental design

No comment

Validity of the findings

No comment

Additional comments

No comment

---

## Round 0.3 · accepted · Accept

· Academic Editor

Accept

Thanks for revising the manuscript. The current version is satisfactory and ready for publication.